# The incidence and severity of COVID-19 in adult professional soccer players in Russia

Eduard Bezuglov[1,2,3], Vladimir Khaitin[4,5], Artemii Lazarev[1,2,6], Evgeniy Achkasov[1], Larisa Romanova[7,8], Mikhail Butovskiy[9], Vladimir Khokhlov[10], Maxim Tsyplenko[11], Alexander Linskiy[12], Petr Chetverikov[13], Magomedtagir Sugaipov[14], Arseniy Petrov[15], Oleg Talibov[2,16], Zbigniew Waśkiewicz[1,17]*

1 Department of Sports Medicine and Medical Rehabilitation, Sechenov First Moscow State Medical University (Sechenov University), Moscow, Russia, 2 HighPerformance Sport Laboratory, Moscow Witte University, Moscow, Russia, 3 Sirius University of Science and Technology, Sochi, Russia, 4 FC Zenit, Saint-Petersburg, Russia, 5 Department of Physical Methods of Treatment and Sports Medicine, Pavlov First State Medical University, Saint-Petersburg, Russia, 6 A.I. Burnazyan Federal Medical and Biophysical Center, Moscow, Russia, 7 Head of Center for Hygiene and Epidemiology of the Federal Medical Biological Agency, Moscow, Russian Federation, 8 Academy of Postgraduate Education under the Federal State Budgetary Unit "Federal Scientific and Clinical Center for Specialized Medical Assistance and Medical Technologies of the Federal Medical Biological Agency," Moscow, Russian Federation, 9 FC Rubin, Kazan, Russia, 10 FC Rostov, Rostov-on-Don, Russia, 11 FC Tambov, Tambov, Russia, 12 FC Sochi, Sochi, Russia, 13 FC Orenburg, Orenburg, Russia, 14 FC Akhmat, Grozniy, Russia, 15 University of Göttingen, Göttingen, Germany, 16 Moscow State University of Medicine and Dentistry, Moscow, Russia, 17 Institute of Sport Sciences, Jerzy Kukuczka Academy of Physical Education, Katowice, Poland

* z.waskiewicz@awf.katowice.pl

**Data Availability Statement:** The raw datasets are uploaded as supplementary .xls file

**Funding:** The author(s) received no specific funding for this work.

## Abstract

There are little data on the incidence, and clinical course of COVID-19 among professional soccer players, and the studies examining putative complications of COVID-19 infections are probabilistic. On February 28, the WHO raised the COVID-19 threat assessment to its highest level. The COVID-19 outbreak became a significant challenge for world health. Around 30 million people got infected with COVID-19 since the beginning of this year. More than 900.000 decease. Thus, examining the incidence of COVID-19 and various aspects of its clinical course in a group of adult professional soccer players would be of great practical interest. The incidence, clinical practice, and severity of COVID-19 infection, as well as the duration of treatment and return to play was studied based on a survey of team physicians and medical records assessment in the group of adult professional soccer players representing the clubs of the Russian Premier-League (RPL) during the period of championship resumption from 01.04.2020 until 20.09.2020. COVID-19 infection was detected in 103 soccer players during COVID-19 screening. This number comprises 14.5% of all soccer players on the rosters of RPL soccer teams and is subjected to regular COVID-19 testing. The asymptomatic course was observed in 43.7% of cases (n = 45). These players were isolated, and their clinical condition was monitored closely. In 56.3% of patients (n = 58), fatigue, headache, fever, and anosmia were the most common symptoms. COVID-19 infection was commonly diagnosed among adult professional soccer players in Russia. However, most cases had a mild course and did not impair return to regular exercise. Only two players were hospitalized with lung lesions and returned to regular sports.

**Competing interests:** The authors have declared that no competing interests exist.

## Introduction

Although their efficacy and safety are dubious, various therapeutic approaches for COVID-19 treatment are utilized. There have been very few studies to date that have investigated the clinical course of COVID-19 infection and its impact on the performance of soccer players [1, 2]. COVID-19 complications such as pulmonary fibrosis cardiac and hepatic consequences are actively studied [3]. The course of COVID-19 infection and its impact on athletic performance has not been studied in professional athletes. Nearly all soccer events had been canceled in March 2020 due to the fast-spreading of the global COVID-19 pandemic. As a result, most soccer players had to stop training [4–6]. During May-June 2020, sports events were resumed in several countries. However, the events took place without fans in most cases.

In all cases when the events were resumed, the organizing sports leagues developed strict prevention and control measures to minimize the risk of infection for the participants [1, 2, 7]. The critical elements of these measures are as follows: close monitoring of the infection rates by PCR- tests; individual face masks and gloves wearing; surface and skin disinfection; adherence to social distancing guidelines. Nevertheless, media reports covering new infection cases among soccer players illustrate that as efficient as they are, these prevention measures cannot completely rule out the possibility of COVID-19 spreading in such a large population as soccer players.

However, there is presently limited data on incidence and clinical course among professional soccer players, and the studies examining putative complications of COVID-19 infections are probabilistic.

This research aims to investigate the clinical course of COVID-19 infection and its impact on the performance of adult professional soccer players as one of the most significant practical importance in sports medicine at the moment.

## Materials and methods

Incidence, clinical course, and severity of COVID-19 infection, as well as the duration of treatment and recovery before return to play, were studied based on a survey of team physicians and medical records assessment in the group of adult professional soccer players representing the Russian Premier-League (RPL) clubs during the period of championship resumption from 01.04.2020 until 20.09.2020.

The data of 710 soccer players who were on the rosters of 16 RPL clubs and 2 National Football League teams (second-tier soccer league in Russia) were included in the analysis. In addition, the study included players who were in the teams' applications for the season and who were regularly tested following the existing national sanitary regulations.

According to the RPL COVID-19 regulation, each player registered for a soccer match must undergo COVID-19 screening by submitting a throat and nostrils swab for a PCR test 3 days before the first play and once weekly after that [6]. Swabs were performed only by specialists from laboratories certified by the Federal Service for surveillance on consumer rights protection "Rospotrebnadzor"—a central Russian governmental entity responsible for sanitary and epidemiological surveillance. Teams used different laboratories. However, all laboratories had state accreditation. Without this test, the special QR-code to access the event would not be issued. All positive test results were automatically submitted to a centralized database and revised in special reference laboratories. Test results were available within 24–48 hours in most cases. According to the Russian quarantine rules, all individuals who tested positive for COVID-19 have to be isolated for 14 days regardless of their clinical symptoms. The quarantine can be lifted only after receiving two negative PCR-test results performed within 24 hours.

Medical records of athletes diagnosed with COVID-19 by a PCR test were studied. Disease course (symptomatic/asymptomatic), the frequency of pulmonary involvement, and the severity of pulmonary lesions were assessed. The prevalence of distinct clinical findings, the therapeutic approaches, the duration of treatment, and recovery before return to play were included in the analysis. The team physicians were interviewed by telephone using a questionnaire compiled in advance. An independent clinical pharmacologist validated the questionnaire. All necessary information was obtained from the team physicians and, in some situations, from the athletes themselves. They measured their temperature and oxygen saturation daily to monitor asymptomatic players during quarantine. Also, they had to report the onset of any of the symptoms that appeared: cough, shortness of breath, loss of smell or taste, sore throat, and headache. This often happened when they were outside the team (at home or in the room at the training base). In such situations, the player was immediately advised to isolate himself.

An independent clinical pharmacologist assessed all the medical records. All radiographic findings were initially evaluated by radiologists experienced in characterizing lung lesions. In addition, all players underwent regular testing following the adopted sanitary regulations, making it possible to identify and timely isolate initially asymptomatic carriers. However, such measures were not taken in Russia concerning representatives of the general population.

The database was created with Microsoft Excel software; statistical analysis was performed utilizing the IBM SPSS 23.0 (Armonk, USA). Continuous data were tested for normality of distribution with the Kolmogorov-Smirnov test. Normally distributed data were described with Mean (M) and standard deviation (SD). Median (Me) and quartiles were used in case of abnormal distribution). Percentage and absolute numbers were provided for categorical data. Mann-Whitney U-test was performed to compare the duration of treatment and recovery before return to play between athletes with and without pulmonary lesions. Spearman's correlation was used for non-normal distributed data. Results were considered statistically significant at $p < 0.05$.

The study was approved by the local ethics committee (Sechenov University, protocol № 30–20 from 21.10.2020). Players provided written informed consent to participate in the survey. They explained that their medical documentation only served scientific purposes, and their data would be protected.

## Results

COVID-19 infection was detected in 103 soccer players in the course of COVID-19 screening (average age: 25,1 ± 4,3 years, height: 183,7 ± 6,3 cm, weight: 76,6 ± 7,0 kg, BMI: 22,7 ± 1,4). This number comprised 14.5% of all soccer players on the rosters and underwent regular PCR testing. Out of the 103 infected players defensive midfielders were most frequently infected (n = 42; 40.8%), followed by defenders (n = 30; 29.1%), strikers (n = 19; 18.4%) and goalkeepers (n = 12; 11.7%). The asymptomatic course was observed in 43.7% of cases (n = 45). These players were isolated, and their clinical condition was monitored closely. Clinical symptoms were observed in 56.3% of cases (n = 58), the most common symptoms being fatigue, headache, fever, and anosmia (Table 1).

In all COVID19 PCR-positive cases lung CT scan was performed. Pulmonary lesions were detected in 36.2% (n = 21) of symptomatic soccer players. And in 23.3% (n = 24) of players with positive test results (in 3 cases, pulmonary lesions were revealed in asymptomatic players. In asymptomatic patients, "frosted glass opacities" extended less than 10% of the lung. There were no fundamental differences in the structure of these players' radiological changes compared to the radiological data of football players with clinical symptoms. All three

**Table 1. Clinical symptoms in soccer players with COVID-19 infection.**

| Symptom | Frequency in symptomatic soccer players |
|---|---|
| | (%, n) |
| Fatigue | 72.4%, 42 |
| Headache | 65.5%, 38 |
| t ≥38˚C | 44.8%, 26 |
| (Anosmia/parosmia) | 41.3%, 24 |
| Sore throat | 31%, 18 |
| 37˚C<T <38˚C | 27.6%, 16 |
| Cough | 15.5%, 9 |
| Diarrhea | 8.6%, 5 |
| Myalgia | 6.9%, 4 |
| Dyspnea | 1.7%, 1 |

asymptomatic players underwent CT because of their willingness, which undoubtedly did not match the current clinical recommendations.

In all 24 cases, pulmonary lesion size was derived from CT images. Less than 10% of lung parenchyma were involved in 70.9% (n = 17) of cases, 11–20% in 16.6% (n = 4), and 20–29% in 8.3% (n = 2) of cases. Lesions involving more than 30% of lung parenchyma were detected only once (4.2%; n = 1). Two players (20 and 36 years old) were hospitalized with lung lesions (26% and 32%). They did not use oxygen therapy during their hospital stay. Lesions were most commonly associated with the following symptoms: fatigue (76.2%, n = 16), headache (71.4%, n = 15), t ≥ 38˚C (52.4%, n = 11). Medical records (n = 61) were available for assessment. Players were clustered by the drug received (Table 2).

The average treatment duration of symptomatic soccer players without pulmonary manifestations of the disease was 14,4 ± 4,8 days, and 17,4 ± 3,5 days in those with pulmonary lesions (p = 0.0083). Pulmonary lesions were not identified as a risk factor contributing to a longer duration of treatment, which were defined as the time for the disappearance of all symptoms (logistic regression, p = 0.09) or delaying the resumption of training as a regular player in the respective soccer team (logistic regression, p = 0.17).

The significant duration of players with and without pulmonary lesions was registered (p = 0.022, Mann-Whitney), and no significant difference has been observed in the course of recovery before return to play (period of return to regular training activity, which was defined as the period from the moment of diagnosis of the disease to the start of training specific to the sport) (p = 0.29, Mann-Whitney). Treatment duration was significantly longer in the players with pulmonary lesions. There were no complications that required oxygen support or pulmonary ventilation. Low-flow oxygen support was not used in any case.

**Table 2. Drugs utilized for the treatment of COVID-19 infection.**

| Drugs | Prevalence (%, n) |
|---|---|
| Antivirals (Umifenovir) | 85.2%, 52 |
| Antibiotics (Azithromycin) | 83.6%, 51 |
| Polivitamins | 59%, 36 |
| Anticoagulants (Enoxaparin) | 26.2%, 16 |
| Interferons (intranasal interferon-gamma) | 14.8%, 9 |
| Anti-malaria agents (hydroxychloroquine) | 11.5%, 7 |
| NSAIDs (COX-2 selective inhibitors) | 6.6%, 4 |

After recovery, before return to training, all players underwent a thorough medical examination which is mandatory in Russia according to the Ministry of Health's particular order. Admission of any football players to participate in competitions and regular training occurs only after passing a mandatory medical examination, performed twice a year. This examination includes a routine physical examination, clinical blood and urine test, ECG (rest and stress-test), spirography, chest X-ray. During this medical examination, no cardiovascular, pulmonary and hematological abnormalities were detected in players who recovered from COVID-19 infection.

Before return to play, the average recovery duration was 18.0 ± 5.0 days (median±IQR). The recovery period lasted 18.0 ± 8.0 days (non-normal distribution) in those who suffered COVID pneumonia and 18.0 ± 5.0 days (non-normal distribution) in those with no pulmonary manifestations. No significant correlation in COVID-pneumonia and the duration of the recovery period was detected (Mann-Whitney, $p = 0.32$). The number of clinical findings assessed the disease severity. In all cases, the COVID-19 disease was mild, and the treatment algorithm in every case was determined individually. The age and clinical findings distribution are non-normal; thus, Spearman's rank correlation coefficient was utilized. There was no significant and robust correlation between the age and clinical findings ($p = 0.1$; $R = 0.18$). Almost all soccer players who had symptomatic COVID-19 disease had already participated in soccer events to the moment of study recruitment.

## Discussion

We have demonstrated the predominance of asymptomatic forms of COVID-19 infection among professional soccer players continuously residing in Russia. There were no severe complications. Furthermore, COVID19did not impair return to training after convalescence. This is the first study to investigate the progression of COVID-19 in professional soccer players with a focus on clinical and radiological findings.

Grant et al. performed a systematic review and a meta-analysis of 148 scientific papers, including a total of 24410 adult patients with COVID-19 disease (age >16 years, average age 49 years) from 9 countries. The most common symptoms described were fever– 78%, cough– 57%, and fatigue– 31%. In addition, the headache was observed in 13% of cases, hyposmia– 25%, and myalgia—17%.

In contrast to our study, Grant et al. assessed all symptoms in patients with a positive test result. Thus, the symptomatic course of the disease was less commonly observed in our study, which is most probably due to the younger age and excellent health of the soccer players [8].

Bergheim et al. performed a retrospective study of 121 symptomatic COVID-19 patients from China (average age 45±15.6 years). Pulmonary lesions were observed in 78% of patients. In our study, only 20.4% of soccer players had pulmonary manifestations, probably due to timely diagnosis and early treatment [9]. Meyer et al. [7] demonstrated a relatively low incidence of SARS-CoV-2 infections among German soccer players. During the study period from May to July 2020 (9 weeks), eight players were tested positive in the first PCR-testing round before the resumption of training sessions. Two players were tested positive in the third round of PCR testing, and 22 remained seropositive throughout the whole season. The numbers reported in this study are much lower than ours. Several factors might explain the differences.

When club training and competitive matches in Russia were resumed, the fans were allowed to enter the stadium, facilitating the spread of infection. According to the sanitary regulations, players could not intersect with the spectators in any way since they immediately got out of the bus into the "clean" zone, which was the under-stands room and the field and the locker rooms. All people (except players) who were in the "clean zone" could get there only by

passing a negative PCR test within 72 hours before the start of the match. In contrast to Russia, viewers were not allowed at Bundesliga games [7].

Another factor influencing the frequency of COVID-19 infections among Russian soccer players could be the frequent flying, thus a large number of close contacts. Each team had at least 20 flights, lasting >2 hours during the study period. Although the teams used charter flights, the risk of infection was nevertheless high. During teams' stay at the training grounds and from the bus to the "clean" zone, all the players must wear masks. But unfortunately, not all

Teams lived separately at the hotels or training grounds, in connection with which the players often stayed overnight at home. However, a higher detection rate in Russian players could be due to meticulous PCR-testing done by the physicians' staff once a week, which allowed them to detect infections early and promptly isolate and treat the tested positive players. An essential aspect of the professional athlete's treatment is strict adherence to anti-doping regulations. If prohibited substance use is necessary for t an athlete has to apply for a Therapeutic Exemption (TUE) by the respective national anti-doping association. Considering that the uniform COVID-19 treatment method has not been established yet, in some cases, it could be problematic to receive a TUE from the WADA retroactively, e.g., after being treated (e.g., with dexamethasone [10]. However, the treatment protocols applied to soccer players did not require the use of forbidden substances such as systemic glucocorticoid use.

The recovery period to enable the safe resumption of training is critically important for soccer players. It is assumed that COVID-19 itself, as well as various therapeutics to treat the infection, might cause adverse effects on organs, primarily on the cardio-respiratory system [11]. However, the side effects might be sub-clinical and not appear instantly [12]. The importance of health monitoring in athletes who recovered from COVID-19 was stressed in previous studies [13].

According to Russian regulations, all professional athletes must undergo compulsory thorough medical examination twice a year. Those athletes who are non-compliant with this requirement will not receive permission from the national sports federation to train and participate in competitions [14]. The presence of a mandatory additional examination before admitting an athlete to physical and sports activity will allow identifying various deviations in the health status of athletes after suffering from COVID. After full recovery, all soccer players who had COVID19 diagnosed passed this type of examination, and none of them was diagnosed with any pulmonary or cardiac pathologies or exercise intolerance. Furthermore, no athletes got advice to restrain from physical activity irrespective of its intensity.

Thus, asymptomatic and symptomatic soccer players with mild pulmonary manifestations of the disease did not demonstrate any impairment in respiratory and cardiovascular function or Exercise intolerance in the short-term after recovery from COVID-19 infection. The long-term effects of COVID-19 should be the focus of future research.

At the survey time, the predominant virus strain was a wild strain (Wuhan strain). Nowadays, the new dominant strain may be more virulent and aggressive (e.g., B.1.167.2). This aspect must be considered when reading the article since all the results were obtained in a specific period when a particular virus strain prevailed.

## Limitations of research

The research design was a retrospective survey that implies possible bias (e.g., recall bias).

## Conclusion

COVID-19 infection was commonly diagnosed among adult professional soccer players in Russia. However, most infections had a mild course and did not impair return to regular exercise.

## Supporting information

**S1 File.**
(XLSX)

## Author Contributions

**Conceptualization:** Eduard Bezuglov, Vladimir Khaitin, Artemii Lazarev, Larisa Romanova.

**Data curation:** Eduard Bezuglov, Vladimir Khaitin, Larisa Romanova, Vladimir Khokhlov, Maxim Tsyplenko, Alexander Linskiy, Petr Chetverikov, Magomedtagir Sugaipov, Oleg Talibov.

**Formal analysis:** Eduard Bezuglov, Vladimir Khaitin, Larisa Romanova, Vladimir Khokhlov, Maxim Tsyplenko, Alexander Linskiy, Petr Chetverikov, Magomedtagir Sugaipov, Oleg Talibov.

**Investigation:** Oleg Talibov.

**Project administration:** Artemii Lazarev.

**Resources:** Vladimir Khaitin, Artemii Lazarev.

**Supervision:** Artemii Lazarev, Evgeniy Achkasov.

**Visualization:** Mikhail Butovskiy, Arseniy Petrov.

**Writing – original draft:** Eduard Bezuglov, Vladimir Khaitin, Artemii Lazarev, Evgeniy Achkasov, Mikhail Butovskiy, Arseniy Petrov, Zbigniew Waśkiewicz.

**Writing – review & editing:** Eduard Bezuglov, Vladimir Khaitin, Artemii Lazarev, Evgeniy Achkasov, Mikhail Butovskiy, Arseniy Petrov, Zbigniew Waśkiewicz.

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
