## [Decision Letter · Decision Letter 0]

20 Dec 2021

PONE-D-21-35577The Incidence and Severity of COVID-19 in Adult Professional Soccer PlayersPLOS ONE

Dear Mr Zbigniew Waśkiewicz,

Indeed on behalf of the journal, we, are grateful to you for submitting your manuscript to PLOS ONE. At outset, I would like to congratulate authors for raising the issue in context through manuscript submission for wider audience reach post publication if considered.   After careful consideration, we feel that it has merit but partially meet PLOS ONE’s publication criteria as it currently stands. Therefore, we invite you to submit a revised version of the manuscript that addresses the points raised during the review process as discussed below. I hope and wish this reviewer's suggestions will be considered as constructive one in improvement  of the manuscript, please.

In context of the above, you are humbly requested to pursue/divulge with corrections/queries as suggested by reviewer two  , so that the scientific validity of  the manuscript submitted can be enhanced. As other two reviewers had already submitted their decision as 'accept' , revised version  submitted would only further strengthen the process  as per the publication criteria of the journal. 

We look forward to receiving your revised manuscript.

Kind regards,

Dr Gopal Ashish Sharma, MBBS, MD

Academic Editor

PLOS ONE

Journal Requirements:

● A clean copy of the edited manuscript (uploaded as the new *manuscript* file).

4. Please amend the manuscript submission data (via Edit Submission) to include author Oleg Talibov.

7. Please include your tables as part of your main manuscript and remove the individual files. Please note that supplementary tables (should remain/ be uploaded) as separate "supporting information" files.

Additional Editor Comments:

Additionally, please look into typos particularly in results and discussions section. Please  indicate denominator in table 2 . 

Reviewers' comments:

Reviewer's Responses to Questions

**Comments to the Author**

1. Is the manuscript technically sound, and do the data support the conclusions?

Reviewer #1: Yes

Reviewer #2: Partly

Reviewer #3: Yes

2. Has the statistical analysis been performed appropriately and rigorously? 

Reviewer #1: Yes

Reviewer #2: Yes

Reviewer #3: Yes

3. Have the authors made all data underlying the findings in their manuscript fully available?

Reviewer #1: Yes

Reviewer #2: Yes

Reviewer #3: Yes

4. Is the manuscript presented in an intelligible fashion and written in standard English?

Reviewer #1: Yes

Reviewer #2: Yes

Reviewer #3: Yes

5. Review Comments to the Author

Reviewer #1: This is a relevant article in the context of the pandemic, because football matches and sports were suspended for many months. Methodology is sound and design is appropriate to estimate incidence. Paper has policy implications also because it highlights the need for regular screening based on the high proportion of asymptomatic cases.

Reviewer #2: Review summary of the article PONE-D-21-355

A. Summary of the research:

The article has addressed incidence, clinical course, severity of infection, types of conservative management and recovery from COVID-19 infection among 710 professional soccer players over 18 league teams in Russia.

It was mostly a study of secondary data analysis besides telephonic interview of team physicians and sometimes from athletes themselves.

Among the players, 14.5% were detected to have COVID-19 infection through screening tests and of them, 56.3% were having varied types of symptoms and 36.2% had pulmonary lesions in their CT scan.

For the purpose of treatment, antivirals, antibiotics, anticoagulants, interferons etc. were used.

Average treatment duration was found as significantly higher among those with pulmonary lesions though there was no significant difference in time required to return to play ground.

The authors have cited 14 different studies as reference and have aptly mentioned that there is paucity of research in this particular field till now.

The main strength of the research lies in exploration of a much less visited area which carries its own significance as soccer players are country representatives and their management too is challenging in regard that none could be administered with systemic glucocorticoids in order largely to adhere to anti-doping regulations.

One limitation as mentioned by the authors as recall bias. Otherwise as nowhere it has been mentioned how many league teams are there in Russia, existence of generalizability cannot be assessed.

B. Examples and Evidences:

1. Major issues (according to the progress of manuscript)

In the introduction part, it is necessary to mention problem of the extent and rationale of the study in the backdrop of the objectives of the study.

In the current study introduction part has been started and ended mentioning few studies in this issue which can be omitted at least at the beginning part.

Objectives of the present study needed to mention at the last paragraph of the introduction.

In “materials and methods” it has been mentioned that team physicians were interviewed over telephone. Whether the interview guide was structured or not, if it was validated or not to be mentioned briefly.

In result section, page no. 7, “admission of any of the ………..with gas analysis” to be discussed very briefly as it is supportive information to build the result but it is not study finding directly.

In the discussion part result of the current study will not be repeated.

In reference part, if no. of authors >3, et al. to be mentioned.

2. Minor issues:

Introductory statement of the abstract part better to describe the problem statement in brief.

Page no. 10, “according to Russian…………competition”, the relevance of this statement to the current study is not clear.

The 2nd limitation, “predominant virus strain………” is questionable/need clarity as a study limitation.

Reviewer #3: a good article , what would be valuable ,if author add a rational why this particular group could be at higher risk or as they may have a good life style the covid -19 condition may have les serious course among this group

6. PLOS authors have the option to publish the peer review history of their article (what does this mean?). If published, this will include your full peer review and any attached files.

Reviewer #1: **Yes: **Pillaveetil Sathyadas Indu

Reviewer #2: **Yes: **Dr. Satabdi Mitra, Assistant Professor, Community Medicine, KPC Medical College and Hospital, West Bengal, India

Reviewer #3: **Yes: **Dr.Noora Alkubaisi

---

## [Author Response · Author response to Decision Letter 0]

31 Jan 2022

Reviewer 1

Thank you very much for your acceptance.

Reviewer 2

Reviewer #1: This is a relevant article in the context of the pandemic, because football matches and sports were suspended for many months. Methodology is sound and design is appropriate to estimate incidence. Paper has policy implications also because it highlights the need for regular screening based on the high proportion of asymptomatic cases.

Answer: good point, no comments.

Reviewer #2: Review summary of the article PONE-D-21-355

A. Summary of the research:

1. Major issues (according to the progress of manuscript)

In the introduction part, it is necessary to mention the extent and rationale of the study in the backdrop of the objectives of the study.

Answer: agree, we put this in the text.

In the current study introduction part has been started and ended mentioning few studies in this issue which can be omitted at least at the beginning part.

Answer: agree, Various therapeutic approaches for COVID-19 treatment are utilized, although their efficacy and safety are dubious.

Objectives of the present study needed to mention at the last paragraph of the introduction.

Answer: agree, we put this in the text.

The objective of this research is to investigate the clinical course of COVID-19 infection and its impact on the performance of adult professional soccer players as one of the biggest practical importance in sports medicine at the moment.

In the result section, page no. 7, “admission of any of the ………..with gas analysis” to be discussed very briefly as it is supportive information to build the result but it is not study finding directly.

Answer: agree, we did it shorter.

Admission of any of the football players to participate in competitions and regular training occurs only after passing mandatory medical examination, which should be performed twice a year. This examination includes routine physical examination, clinical blood and urine test, ECG (rest and stress-test), spirography, chest X- ray.

In the discussion part the result of the current study will not be repeated.

Answer: agree, deleted.

In reference part, if no. of authors >3, et al. to be mentioned.

Answer: agree, done.

2. Minor issues:

Introductory statement of the abstract part better to describe the problem statement in brief.

Answer: agree, we have added.

Page no. 10, “according to Russian…………competition”, the relevance of this statement to the current study is not clear.

Answer: The presence of a mandatory additional examination before admitting an athlete to physical and sports activity will allow to identify various deviations in the health status of athletes after suffering from COVID.

The 2nd limitation, “predominant virus strain………” is questionable/need clarity as a study limitation.

Answer: agree, Sure, it is not a limitation of the research. But this aspect must be taken into account when reading the article, since all the results were obtained in a specific period of time when a specific strain of the virus prevailed.

---

## [Decision Letter · Decision Letter 1]

21 Feb 2022

The Incidence and Severity of COVID-19 in Adult Professional Soccer Players

PONE-D-21-35577R1

Dear  Prof Zbigniew Waśkiewicz ,

We’re pleased to inform you that your manuscript has been judged scientifically suitable for publication and will be formally accepted for publication once it meets all outstanding technical requirements.

Kind regards,

Dr Gopal Ashish Sharma,

MBBS, MD

Academic Editor

PLOS ONE

Additional Editor Comments (optional):

Reviewers' comments:

Reviewer's Responses to Questions

**Comments to the Author**

1. If the authors have adequately addressed your comments raised in a previous round of review and you feel that this manuscript is now acceptable for publication, you may indicate that here to bypass the “Comments to the Author” section, enter your conflict of interest statement in the “Confidential to Editor” section, and submit your "Accept" recommendation.

Reviewer #2: All comments have been addressed

2. Is the manuscript technically sound, and do the data support the conclusions?

Reviewer #2: Yes

3. Has the statistical analysis been performed appropriately and rigorously? 

Reviewer #2: Yes

4. Have the authors made all data underlying the findings in their manuscript fully available?

Reviewer #2: Yes

5. Is the manuscript presented in an intelligible fashion and written in standard English?

Reviewer #2: Yes

6. Review Comments to the Author

Reviewer #2: The article is well written and has aptly addressed a very important issue in current context. All the issues identified have been discussed and modified appropriately. Writing style is also very good.

7. PLOS authors have the option to publish the peer review history of their article (what does this mean?). If published, this will include your full peer review and any attached files.

Reviewer #2: **Yes: **Dr Satabdi Mitra

---

## [Editor Report · Acceptance letter]

27 May 2022

PONE-D-21-35577R1 

The Incidence and Severity of COVID-19 in Adult Professional Soccer Players in Russia 

Dear Dr. Waśkiewicz:

I'm pleased to inform you that your manuscript has been deemed suitable for publication in PLOS ONE. Congratulations! Your manuscript is now with our production department. 

Kind regards, 

on behalf of

Dr. Gopal Ashish Sharma 

Academic Editor

PLOS ONE